# Optimizing Energy Consumption: A Case Study of LVDC Nanogrid Implementation in Tertiary Buildings on La Réunion Island

Olivia Graillet [1,2,*] , Denis Genon-Catalot [3], Pierre-Olivier Lucas de Peslouan [1], Flavien Bernard [1,2], Frédéric Alicalapa [1] , Laurent Lemaitre [2] and Jean-Pierre Chabriat [1]

1   ENERGY-Lab., University of La Réunion, 97490 Saint-Denis, France
2   Intégrale Ingénierie, 97435 Saint-Gilles-les-Hauts, France
3   LCIS, Grenoble INP, University of Grenoble Alpes, 26000 Valence, France
*   Correspondence: olivia.bory-devisme@univ-reunion.fr

**Abstract:** In the context of an insulated area with a subtropical climate, such as La Réunion island, it is crucial to reduce the energy consumption of buildings and develop local renewable energy sources to achieve energy autonomy. Direct current (DC) nanogrids could facilitate this by reducing the energy conversion steps, especially for solar energy. This article presents the deployment and efficiency evaluation of a 48 VDC low-voltage direct current (LVDC) nanogrid, from conception to real-world installation within a company. The nanogrid consists of a photovoltaic power plant, a lithium–iron–phosphate (LFP) battery, and DC end-use equipment, such as LED lighting and DC fans, for two individual offices. For identical test conditions, which are at an equivalent cabling distance and with the same final power demand, the total power consumed by the installation is measured for several stages from 50 to 400 W, according to a 100% DC configuration or a conventional DC/AC/DC PV configuration incorporating an inverter and AC/DC converter. The methodology used enables a critical view to be taken of the installation, assessing both its efficiency and its limitations. Energy savings of between 23% and 40% are measured in DC for a power limit identified at 150 W for a distance of 25 m. These results show that it is possible to supply 48 VDC in an innovative way to terminal equipment consuming no more than 100 W, such as lighting and air fans, using the IEEE 802.3 bt power over ethernet (PoE) protocol, while at the same time saving energy. The nanogrid hardware and software infrastructure, the methodology employed for efficiency quantification, and the measurement results are described in the paper.

**Keywords:** LVDC nanogrid; energy efficiency; power over ethernet; solar energy; line losses; building

## 1. Introduction

In the context of global energy transition, it is essential to reduce the carbon impact of buildings, from the design phase through to the operational phase. During operation, the building's carbon impact will first depend on the primary energy source used to supply electricity and secondly on the building's electricity consumption, which, in turn, will depend on the chosen electrical architecture, the efficiency of the terminal equipment, and user behavior.

In response to the challenges about the operational phase, the French government has published a Tertiary Sector decree [1] and a Building Automation and Control System (BACS) decree [2]. The aim of the first decree is to reduce the energy consumption of commercial buildings by 40% by 2030, by 50% by 2040, and by 60% by 2050. The aim of the second decree is to require that buildings have instrumentation to monitor energy savings achieved or enable them to be improved. Another government objective is for non-interconnected zones, including La Réunion island, to achieve energy autonomy by 2030 [3].

La Réunion, which is a French overseas department located in the south-western part of the Indian Ocean, is currently 62% dependent on fossil fuels for electricity production [4].

Renewable energies, which currently represent 38% of the island's annual electricity production [4] (more than 20% for hydro power and less than 9% for solar energy), could represent a solution for decarbonizing electricity production. As a matter of fact, the local annual solar radiation is between 1100 and 2100 kWh/m$^2$, and the territory, which is isolated from any external other grid, can experience cyclonic phases, including power cuts. Reducing the energy consumption of buildings, which account for 86% of annual electricity consumption [5] of La Réunion, and promoting the development of local renewable energies, such as solar energy, appear to be a lever of action to contribute to the energy autonomy of the department. It is amenable to a systemic approach, which takes into account the specific features of a location. However, the major challenge of renewable energies is their intermittency. The results of this study [6] show that the optimization of microgrid systems based on probabilistic models enables the effective management of renewable energies, thereby helping to reduce the carbon impact by promoting the more sustainable and efficient use of energy resources.

Micro- and nanogrids, which are small-scale electrical grids, are currently highlighted due to their ability to facilitate renewable energies penetration. They include one or more energy sources, often renewable, a storage system, and an intelligent energy management system. They can operate connected to the grid or in island mode, disconnected from the grid [7]. Microgrids could be particularly well suited to an island such as La Réunion, which needs to achieve energy autonomy.

End-to-end direct current (DC) microgrids use direct current electricity from the production to the final equipment. They have the distinct advantages of reducing the need for power converters and shortening energy transmission distances. They are considered to be one of the primary solutions for energy savings according to several studies [7–17]. There are even communities, such as those in the USA, that live comfortably and daily on 100% DC microgrids [18]. Despite these promising estimates, a major challenge lies ahead for the industrial sector, as electrical equipment manufacturing norms and standards need to evolve if DC installations are to be democratized on a large scale [19]. The costs of DC circuit breakers and appliances are currently still high. Additional studies and feedback are necessary to guide legislative and industrial developments. Otherwise, concerning the scientific challenges, the number of experimental sites needs to grow, to confirm or otherwise the estimates predominantly based on simulations. The results require real data to validate or refute hypotheses regarding the efficiency of micro- and nanogrids [19–22].

As points of comparison in the department of La Réunion, microgrids have already been deployed in isolated sites, such as the Cirque of Mafate [23,24]. The houses located on this site have the particular characteristic of being completely disconnected from the electrical grid. Each maintenance intervention on the equipment requires helicopter transportation, which has an impact in terms of cost and carbon impact. Finding sustainable solutions for remote control and strategies aimed at preserving the lifespan of the installed equipment is, therefore, essential.These microgrids utilize solar energy as the primary energy source and lithium-ion batteries for energy storage, or hydrogen storage for hybrid sites [23]. When solar power generation is insufficient, users resort to fuel generators. The electricity is then distributed to households in alternating current (AC) in order to conform to current norms and to be consistent with final common equipment, like washing machines and televisions. As can be seen in the electrical architecture of the hybrid site [23], a 48 VDC bus is used at the output of the battery and the hydrogen system, before being converted to 230 VAC for the buildings. According to [25,26], a 48 VDC distribution bus is the most suitable in terms of costs and efficiency for deployments over short distances in buildings. Given the reduced distances for electricity distribution in microgrids, it seems pertinent, therefore, to experimentally study the advantages and limitations of 48 VDC distribution. Finally, regarding the other microgrid example [24], the contribution is focused on energy management, through a user interface. Energy gains of 22% were measured in the scenario where the user follows the recommendations of the user interface as much as possible. It seems relevant, therefore, to associate energy management with 48 VDC distribution to maximize the performance of a nanogrid.

In the Indian Ocean sector, other microgrids have been deployed and are operational in Madagascar [27]. In their case, the distribution buses used are exclusively in direct current; a 60 VDC bus connects several nanogrids, and within the dwellings, 12, 24, or 48 VDC buses are used depending on the application cases.

Concerning the scientific challenges of nanogrids, according to M. Uddin et al. [28], microgrids are currently facing several technical challenges, such as "operation, component and compatability, integration of distributed energy resources and protection". In this paper, firstly, a contribution regarding the "component and compatibility" challenge is provided. The presented hardware and software architecture allows for experimental feedback on the communication protocols used for the instrumentation and supervision of the DC nanogrid, such as IEEE 802.3bt power over ethernet standard [29]. Secondly, with respect to "integration of distributed energy resources", optimal nanogrid sizing and design is necessary to facilitate integration of renewable energies by reducing the costs and improving efficiency. Different tools are used for nanogrid design [30–32] and some experimental validation tests have been carried out in the laboratory. In this paper, the efficiency of the 48 VDC nanogrid is assessed using a methodology taking into account voltage drops in DC, and the theoretical values are compared on real equipment deployed in a building. The DC measurements are compared to a DC/AC/DC distribution, which corresponds to the current solar energy installations in buildings. The parameters revealed by this experiment are values that could be incorporated into nanogrid sizing tools in order to refine the accuracy of the models.

The article is organized as follows: Section 2 presents the LVDC nanogrid deployment and details the hardware and software choices. Section 3 outlines the methodology used to theoretically evaluate the efficiency of the nanogrid. Section 4 compares the theoretical results with the experimental measurements taken on site. Finally, Section 5 summarizes the conclusions and prospects for this work.

## 2. LVDC Nanogrid Deployment

The building studied is an engineering firm made up of around fifty employees with working hours of between 8 am and 6 pm on average. The building is west-oriented and the surface area of the premises is 590 m$^2$. The energy consumption of the premises is constantly monitored by category of equipment (air conditioning, lighting, ventilation, computer equipment, electric vehicle charging stations, and other sockets).

The premises consumed 83 kWh/m$^2$ in 2022, and the breakdown of energy consumption showed that direct current appliances, which were led lighting, EV charging, and computer equipment, accounted for 42% of the total electricity consumption. This percentage highlights the fact that direct current has a larger share in the tertiary sector than in the residential sector, given the amount of IT equipment. It is, therefore, a lever for optimizing the overall energy reduction of buildings.

Two individual offices were identified on the company premises to serve as demonstrators. The nanogrid distribution rack was located in the technical room, approximately 25 m in cable distance from the two offices. Figure 1 shows the locations for implementation of the nanogrid deployment on the island and in the building.

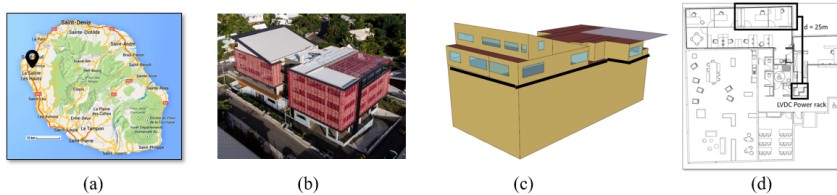

| (a) | (b) | (c) | (d) |

**Figure 1.** (**a**) Localization of the company on the west coast of the island. (**b**) Photo of the building where the nanogrid is deployed. (**c**) Highlighted location of the premises, located on the 3rd floor. (**d**) Localization of the two individual offices used for the DC nanogrid deployment, at 25 m from the power rack.

### 2.1. Hardware Architecture—48 VDC Distribution

The LVDC nanogrid has a 3 kWp photovoltaic power station, which is installed on the sloped roof of the building. In order to correlate the external meteorological condition data with the internal energy production and consumption data and thermal comfort, a solar and meteorological station was also installed on the roof (photograph in Figure 2).

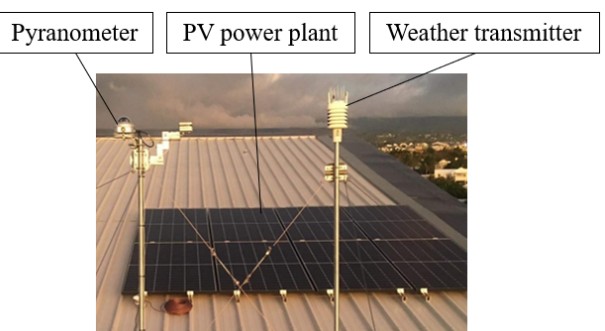

**Figure 2.** Photo of the 3 kWp PV power plant and weather station on the roof.

The output voltage is regulated to 48 VDC by an MPPT (maximum power point tracker) regulator. The 48 VDC distribution bus can then be powered either by the PVs, or by a 2.4 kWh capacity lithium–iron–phosphate battery, or by the grid through an inverter. In the studied case, the grid is not used as a power source, as the nanogrid is totally disconnected from the grid. The inverter is installed to conduct efficiency tests, either in a DC/AC/DC architecture or in a completely end-to-end DC architecture. Figures 3 and 4 show the distribution rack of the DC nanogrid and the block diagram with the components, respectively.

The protective elements are not detailed in the context of this article, but they consist of a DC circuit breaker adapted to the caliber of the equipment. During the measurement phases, these circuit breakers allow for switching on or tripping the various components connected to the 48 VDC bus. The nanogrid is supplemented by 48/24 VDC and 48/12 VDC to power equipment that requires these distribution buses, such as the weather station (24 VDC).

Power over ethernet (PoE) technology is chosen for the energy distribution and the control of the terminal equipment. Indeed, it operates natively at 48 VDC. The main advantage is to use only a single RJ45 cable for electrical power and to provide an internet network to the equipment. The maximum power allowed for the 802.3 bt protocol being 100W, the terminal equipment used in this architecture includes LED panels (max 35 W) and DC fans (max 20 W). As a point of comparison with other LVDC nanogrids, in the "DC nanogrid house" [33] PoE is used to power the indoor environmental sensors.

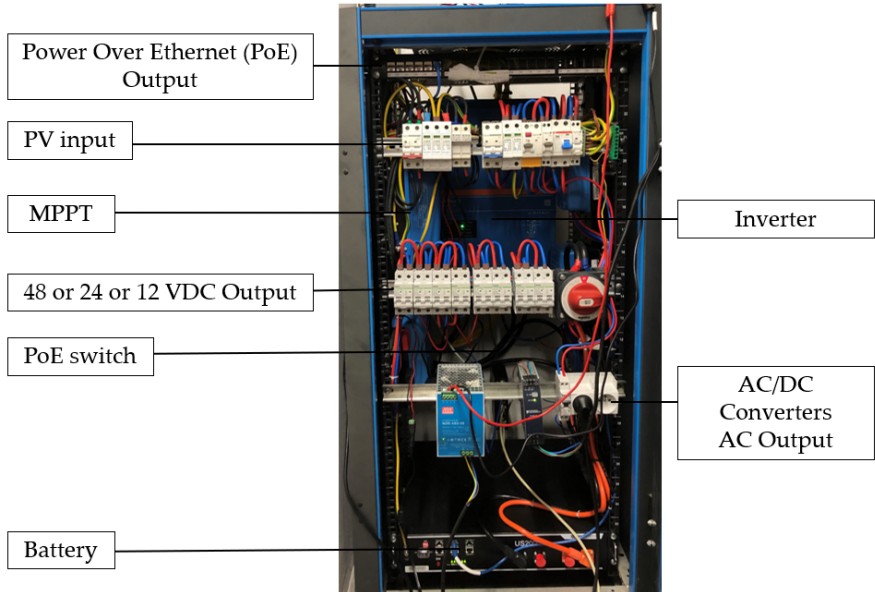

**Figure 3.** Photo of the power rack distribution of the LVDC nanogrid.

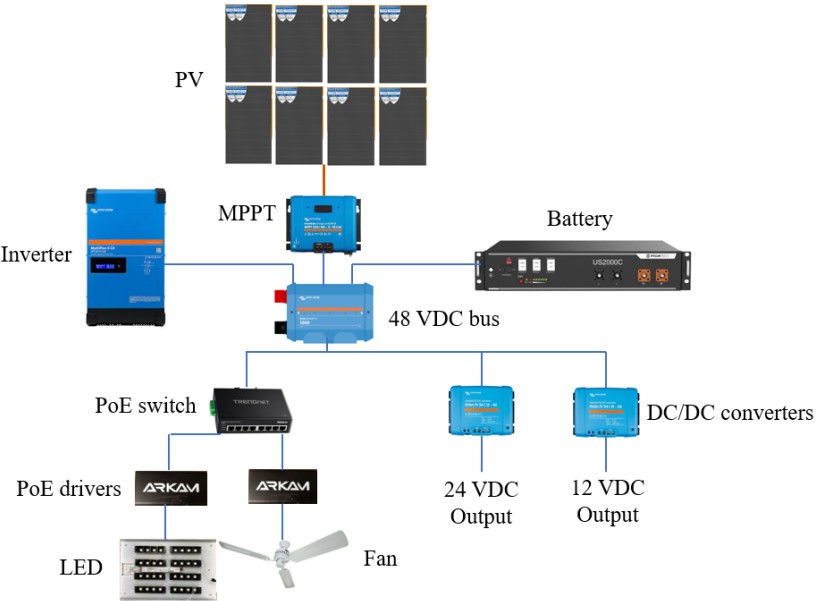

**Figure 4.** Block diagram including the LVDC nanogrid components.

Regarding the operational modes, currently, the nanogrid operates in a fully isolated manner, remaining disconnected from the grid and relying solely on solar energy and battery power. Energy consumption occurs from 8:00 am to 6:00 pm for terminal equipment. Energy management entails a dimming of lighting based on the brightness and user presence, facilitated by additional sensors on the PoE drivers. The comprehensive energy management strategy is currently a developing aspect that is planned within the project framework.

All the other communication protocols used to retrieve the measurements from the nanogrid are detailed in the following section.

*2.2. Software Architecture*

2.2.1. Data Communication

All components of the nanogrid are interconnected. The communication links are depicted in Figure 5.

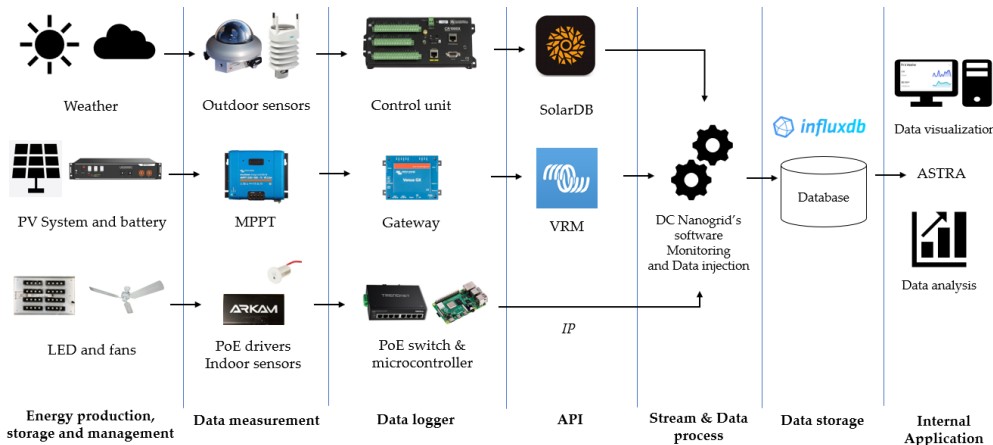

**Figure 5.** Diagram of the data pipeline in the LVDC nanogrid.

With respect to the energy production and storage segment, the MPPT regulator, inverter, and battery are linked via a Ve.bus communication to a gateway, which is connected to the local internet network. The data, such as for the voltage (V), current (A), and power (W), measured and recovered by the gateway are subsequently transmitted to a user interface termed "VRM" provided by the manufacturer.

The solar and meteorological station is integrated into the IOS-net network of stations overseen by the ENERGY-Lab laboratory. As a result, open data are transmitted in near real-time to both a web and mobile application [34].

Both the VRM application and the SolarIO application have an associated application programming interface (API). Information is retrieved using the hypertext transfer protocol (HTTP) communicating with the respective APIs.

For the PoE drivers, each device has an Internet protocol (IP) address. As a result, it is feasible to retrieve voltage (V), current (A), and power (W) information by querying the drivers via secure socket shell (SSH) communication. The PoE drivers have two possible power supply options: either a direct PoE supply or a 48 VDC power input combined with an ethernet input. The driver has three possible outputs: two for the 48 VDC loads, with a maximum of 1A, and one serial output for communication with a sensor. In our setup, a luminosity and presence sensor is connected to the driver, allowing for local lighting regulation based on the user presence and ambient light levels.

2.2.2. Human Machine Interface (HMI)

Currently, the existing instrumentation of electrical equipment within the enterprise is segmented into multiple user interfaces: one for air conditioning, another for general energy consumption, one for lighting management, one for electric vehicle charging stations, another for environmental indoor IoT (Internet of Things) sensors, and a final one for the weather station. Specifically for lighting, dimming is set using the DALI (digital addressable lighting interface) protocol and controlled via a mobile application, but the measurements are not usable. In the case of fan units, the user controls them manually, using remote controls. The entirety of these interfaces or platforms does not allow for data retrieval for subsequent processing. When it does, it is typically in the form of .csv files that can be manually extracted. This procedure requires numerous hours of work for data analysis undertaken by energy management engineers in the company.

In the case of the LVDC nanogrid deployed for this project, the advantage of using electrical equipment connected to the network or equipped with a dedicated API greatly simplifies data processing.

By using PoE technology in particular for certain equipment, the main advantage is that it can supply power and at the same time collect, control, and analyze measurement data from equipment via a single RJ45 cable. Moreover, this facility for measuring energy consumption in tertiary buildings could be a major advantage, as building owners are required to submit their data to the OPERAT [35] platform set up by ADEME (the French environment and energy management agency) in order to comply with the objectives of the tertiary decree. In this LVDC nanogrid, all the gathered data are ultimately aggregated into open-source database technology of the influxdb type. A screenshot of the influxdb-v2 interface implemented is shown in Figure 6.

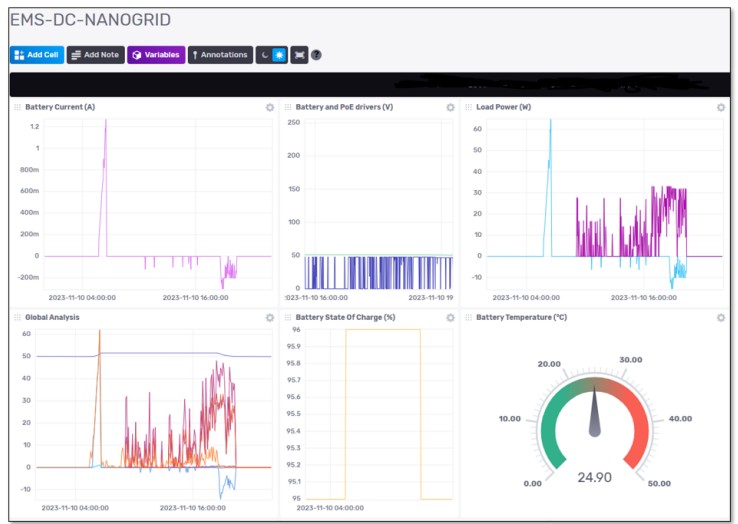

**Figure 6.** InfluxDB monitoring interface of the nanogrid.

On Figure 6, the following informations are displayed:

- In the first graph, the battery charge or discharge current (pink curve).
- In the second graph, the battery voltage (green curve) and the voltage of the PoE LED drivers (purple curve). The latter fluctuates depending on the LED lighting, which is determined by presence or brightness measurements made by the sensor connected to the driver.
- In the third graph, the power consumed or produced by the battery (blue curve) and the power consumed by the PoE drivers (purple curve).
- In the fourth graph, there is an overall analysis that overlays current and voltage measurements regarding the PV panels, the battery, and the drivers.
- The last graph displays the battery state of charge.
- The widget shows the battery temperature for visual monitoring.

To streamline the analysis of the nanogrid's efficiency, calculations pertaining to the quantification of voltage drops and the overall system efficiency are automated and are displayed on an internally developed monitoring interface within the laboratory. This HMI development, shown in Figures 7 and 8, is carried out using a well-known React.js framework called Next.js. This technology offers serverless capabilities to deliver fast and dynamic web pages. Since this HMI interacts with data, it is essential to pay close attention to its data fetching capabilities. In this regard, Next.js provides built-in functions, making it a wise choice for the given tasks. A specific API was developed to query data directly from InfluxDB within Next.js, ensuring rapid data retrieval, which is vital for real-time energy management dashboards.

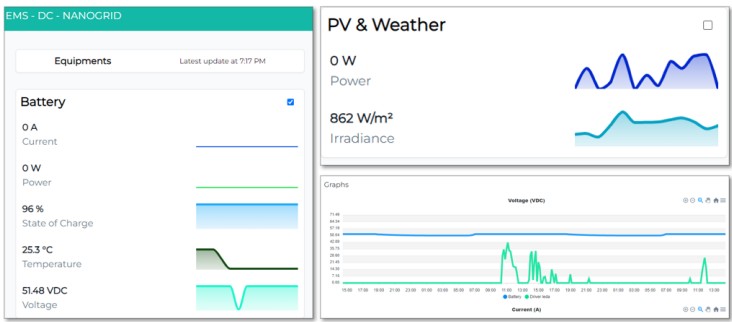

**Figure 7.** Human machine interface—irradiance (W/m²), voltage (V), current (A), and power (W) monitoring.

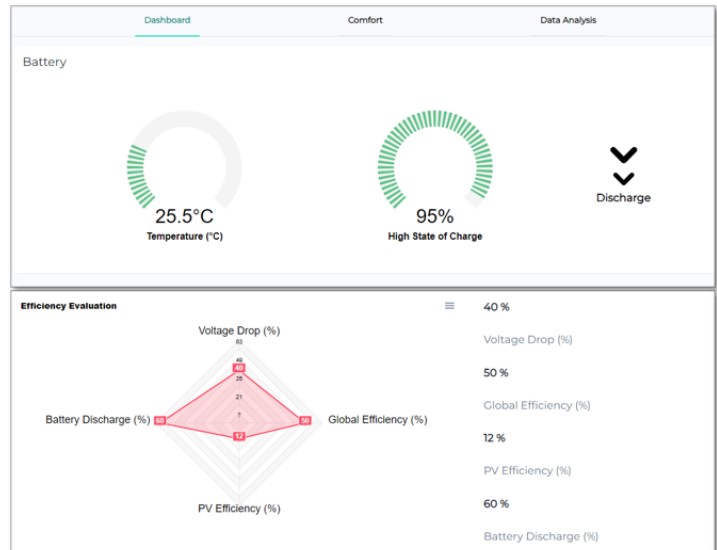

**Figure 8.** Human machine interface—battery monitoring and radar chart model for efficiency evaluation.

The details of the calculations used to determine the system's efficiency are provided in the following section.

## 3. Efficiency Evaluation Methodology

One of the primary constraints in deploying direct current (DC) within buildings pertains to the uncertainty regarding the behavior of DC distribution buses in terms of stability. A significant drop in the distribution voltage can lead to degradation of terminal equipment and cause cable overheating due to the increased current consumption. Regarding the overall efficiency of the system, it would be 100% in the case where the power supplied equals the power consumed. However, joule losses in the cables and the efficiency of the components inevitably affect the system's performance. Consequently, the objective of this LVDC nanogrid is to reduce energy losses during energy distribution, and the instrumentation carried out in this project aims to quantify its advantages and its limitations. The efficiency measurement was carried out under two scenarios on the nanogrid: in one case where the DC/AC and AC/DC converters are integrated into the electrical distribution, and in another case, where the distribution is carried out entirely in 48 VDC. The data are extracted from the supervision interface, which allows quantifying these elements.

### 3.1. Voltage Drop

The methodology from [25] is employed to quantify the theoretical voltage drops that could occur on the DC bus based on the following calculations: First, the resistance $r(\Omega)$

of the cable is calculated, followed by the percentage (%) of voltage drop. This is based on factors such as the distance, cable cross-sectional area, distribution voltage, and the required power:

$$r = \frac{\rho \times 2L}{S} \tag{1}$$

where $\rho$ = 0.01724 ohm, mm$^2$/m is the copper resistivity, $L$ multiplied per 2 for DC distribution is the distance cable (m), and $S$ is the cable cross-section (mm$^2$). Then, the voltage drop $\Delta v$%:

$$\Delta v\% = \frac{100 \times I \times r}{V} \tag{2}$$

with $V$ the DC bus voltage (V) and $I$ the current (A) used, which is calculated from the final power required. As an example case, the evolution of the voltage drop depending on the power load demand is plotted in Figure 9 for two different cable cross-sections: 1.5 mm$^2$ and 2.5 mm$^2$.

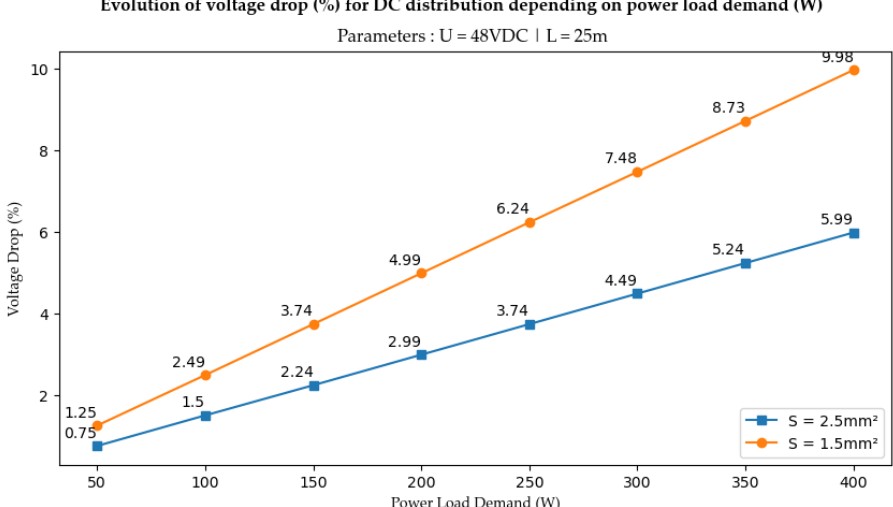

**Figure 9.** Comparison of voltage drops evolution for 1.5 mm$^2$ and 2.5 mm$^2$ cable cross-section.

These two sections are the most commonly used in buildings. The distribution voltage chosen is 48 VDC and the connection distance is 25m. These parameters correspond to the nanogrid deployment described previously. If we refer to the requirements of standard NFC-15100 for low-voltage electrical installations, a voltage drop of less than 3% must be respected for lighting and 5% for other equipment. Even though this standard applies to AC, we consider that we need to respect the same values in order to offer an architecture that complies with current safety standards. From the curves in Figure 9, it can be seen that theoretically the 5% value is expected to be up to 200 W for a 1.5 mm$^2$ cross-section and around 330 W for 2.5 mm$^2$ for a wiring distance of 25 m at 48 VDC. In order to compare this methodology with experimental values, a testbench and the measurement results are presented in Section 4.

In the context of direct current (DC) distribution, as the current consumed by the load increases, the voltage drop also increases. Consequently, the total power consumed rises, impacting the nanogrid's energy efficiency. Regarding an electrical distribution including solar energy in line with today's norms, it transitions from direct current to alternating current (AC) by incorporating an inverter and an AC/DC converter. In this case, it is assumed that the voltage drops are negligible for an identical cable cross-sectional area because at 230 VAC, the current is lower. Therefore, the impact on the nanogrid's energy efficiency in this case will depend on the efficiency of the DC/AC and AC/DC converters.

In order to establish a comparison between these two kinds of deployment, a comparison model is proposed in the next subsection.

*3.2. Efficiency Model*

A theoretical efficiency model is established for this study, based on the following fixed parameters and for different points of power demand $Pn(W)$.

### 3.2.1. Efficiency for DC/AC/DC Distribution

For an assessment of the efficiency of DC/AC/DC distribution, the theoretical system includes a double conversion of current: firstly, from DC (produced by the photovoltaic panels or another DC source) to AC (to comply with the current electrical standard in buildings), and secondly, from AC to DC for all appliances, such as batteries, electronic equipment, and LEDs. This is a representation of what currently happens in buildings that receive solar energy and use direct current appliances. This type of installation comprises an inverter, to transform the direct current from the solar panels into standard 230 VAC, and an AC/DC converter, to supply one or more devices with direct current. This study is based on a single distribution line, so it is assumed that there is only one AC/DC converter to supply an appliance with a power rating of 50 to 400 W. Figure 10 is a schematic representation of this type of installation.

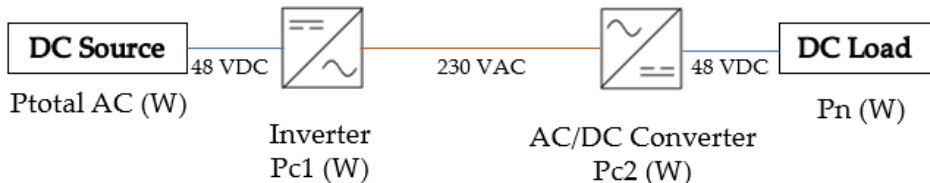

**Figure 10.** Diagram of a DC/AC/DC distribution corresponding to current installations in buildings.

The system efficiency $\eta th_{AC}(\%)$ is calculated in Equation (3) by dividing the end device power consumption $Pn(W)$ by the total power consumed by the system $Ptotal_{AC}(W)$. In Equation (4), $Ptotal_{AC}$ is calculated by taking into account the impact of the converter's energy consumption, given by the datasheet of the components:

$$\eta th_{AC} = \frac{Pn}{Ptotal_{AC}} \times 100 \tag{3}$$

$$Ptotal_{AC} = Pn + Pc1 + Pc2 \tag{4}$$

with $Pc1$ the inverter power consumption (W) and $Pc2$ the AC/DC converter power consumption (W).

For the study of DC/AC efficiency, particularly concerning inverters, formulas and testing protocols have been developed in the United States by the California Energy Comission (CEC) and Sandia National Laboratories [36,37]. In Europe, the same work adapted to the local climate was carried out by the Joint Research Centre (JRC), the scientific and technical research laboratory of the European Union [38]. Their research has notably determined weighting coefficients to be applied across various operating levels of an inverter throughout its entire power cycle. While not utilized within the scope of this study, it is planned to integrate this methodology in future work.

### 3.2.2. Efficiency for DC Distribution

For a DC distribution efficiency evaluation, the theoretical system hypothesis is that 48 VDC voltage is distributed from the source to the final DC appliance, as represented in Figure 11.

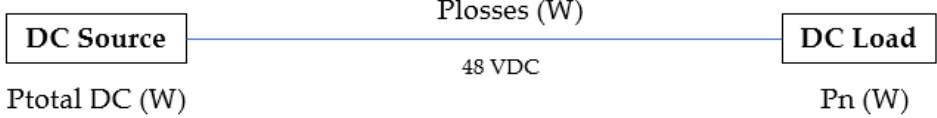

**Figure 11.** Diagram of a DC distribution corresponding to an end-to-end LVDC nanogrid.

The system efficiency $\eta th_{DC}(\%)$ is calculated in Equation (5) by dividing the power demanded by the end device $Pn(W)$ by the total power consumed by the system $Ptotal_{DC}(W)$. In Equation (7), $Ptotal_{DC}$ is calculated by taking into account the impact of the voltage drops, calculated in Equation (6) with $P_{losses}(W)$ as follows:

$$\eta th_{DC} = \frac{Pn}{Ptotal_{DC}} \times 100 \tag{5}$$

$$P_{losses} = (\frac{Pn}{V})^2 \times r \tag{6}$$

$$Ptotal_{DC} = Pn + P_{losses} \tag{7}$$

with $V$ being the DC bus voltage (V), and $r$ the cable resistance ($\Omega$) calculated in Equation (1).

### 3.2.3. Superposition of DC/AC/DC and DC Distribution Efficiency

According to the results obtained by the model which involves calculating $\eta th_{DC}(\%)$ and $\eta th_{AC}(\%)$ in Equations (3) and (5) for different levels of final power demand $Pn(W)$, a meeting limit point is obtained between the efficiency of a DC/AC/DC system and a DC system for two different cable cross-sections. The results obtained for $\eta th_{AC}(\%)$ and $\eta th_{DC}(\%)$ are shown in Figure 12. For 1.5 mm², the maximum power limit is 300 W because beyond that the 48 VDC efficiency becomes lower than AC, and for 2.5 mm², the limit is 400 W. However, it does not seem wise to consider DC deployments that would aim to get closer to these limits, because if we refer to the results obtained in Figure 9, it can be seen in Figure 12 that although the efficiency is proven to be higher in DC under these conditions, the theoretical voltage drops would be outside the norm if we are close to the maximum power limit—higher than 7% for 1.5 mm² and higher than 6% for 2.5 mm².

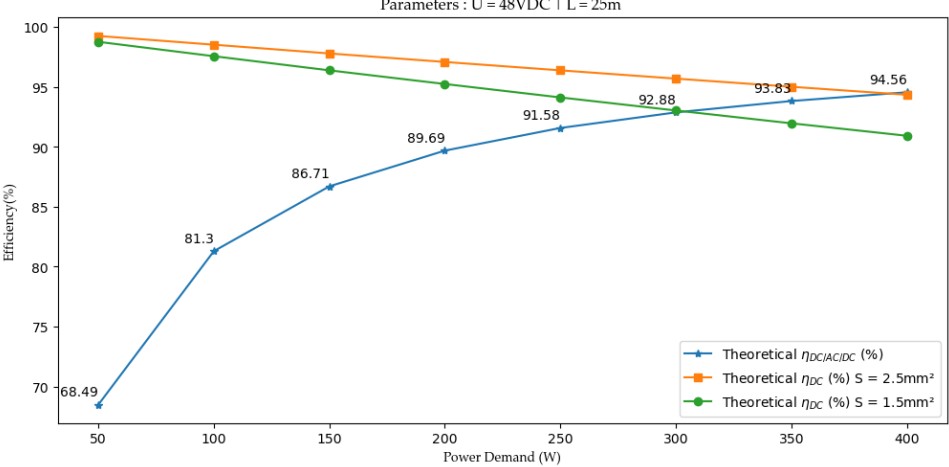

**Figure 12.** Efficiency model AC vs. DC distribution—comparison for 1.5 and 2.5 mm² cable cross-section.

As the aim of this model is to be generic, the input parameters, such as the DC bus voltage or wiring distance, can be modified to give a different meeting point between the

efficiency of the two systems. To illustrate these theoretical values, the following section presents the testbench set up to provide experimental values for the established hypotheses.

## 4. Results and Discussion

### 4.1. Experimental Testbench

To quantify the reliability of the theoretical model, an experimental testbench is set up on the nanogrid, a diagram of which is represented in Figure 13.

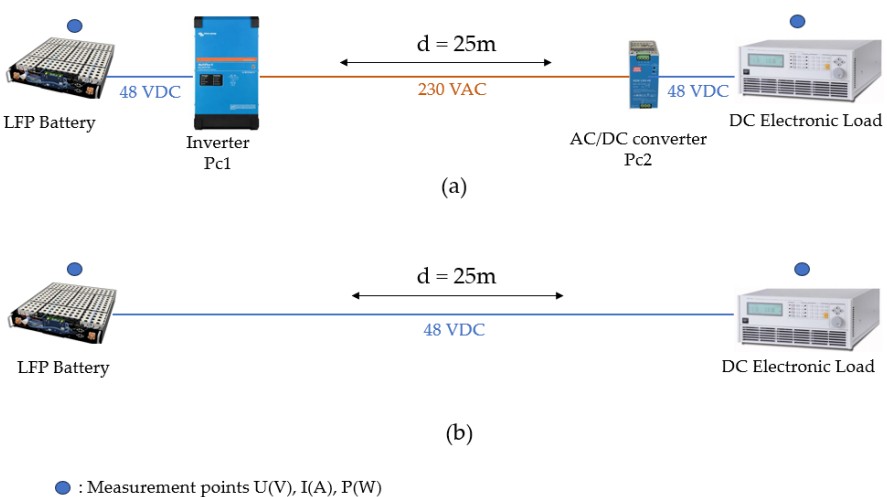

**Figure 13.** Testbench setup for DC/AC/DC distribution (**a**) and for DC distribution (**b**).

The blue dots on the diagram represent the following measurement points: current (A), voltage (V), and power (W). These measurements are carried out by the DC electronic load and the battery. Each has its own internal components for taking these measurements. The battery transmits its data using the Victron API and the programmable load can communicate via a serial link or IP network. As all the equipment is connected as described in Figure 5, the measurement data are entered directly into the database. Measurements were carried out with 3G 1.5 mm$^2$ and 3G 2.5 mm$^2$ cables, each 25 m long, to represent the actual nanogrid deployment on the company's premises. The LFP battery is used as an energy source as solar energy is intermittent, and it would have been inappropriate to compare two measurements taken at different times as the sunshine levels would not have been exactly the same. The DC electronic load is used to simulate a load from 50 to 400 W. All the equipment references and their parameters are listed in Table 1, according to the manufacturer's data. The next sections present the experimental measurements obtained on the testbench.

**Table 1.** List of references and parameters of the components used for the experimental testbench. The power corresponds to the power consumed by the component.

| Equipment | References | Manufacturer | Power (W) |
|:---:|:---:|:---:|:---:|
| Battery | US2000C | Pylontech | - |
| Inverter | Multiplus II | Victron | 18 W |
| AC/DC converter | NDR-240-48 | Meanwell | 5 W |
| DC Electronic load | 63800 | Chroma ATE | 1.8 kW max |

### 4.2. Comparison between DC/AC/DC and 48 VDC Distribution for a Cable Cross-Section of 1.5 mm$^2$

Figure 14a compares the experimental efficiency curves of the DC/AC/DC and DC distributions for a 1.5 mm$^2$ cross-section and a distance of 25 m. From the experimental measurements, it can be seen that the efficiency is 40.8% higher with 48 VDC for a power demand of 50 W than with a DC/AC/DC system. For a load of 100W, the efficiency is 23% higher than in DC/AC/DC. The experimental power limit identified under these conditions

is around 270 W because beyond this point the DC/AC/DC efficiency is higher. It should be noted that the voltage drops measured experimentally ranged from 2.59% for 50 W to 12.8% for 250 W, as shown in Figure 14b. In order to comply with NFC15-100 standards, ref. [39] for example, the final power should not exceed 100 W for these conditions (25 m at 48 VDC), in order to ensure a 5% voltage drop. This result is consistent with a similar study of K. Hafsi et al. [40] that measured the 802.3 bt standard—for a distance of 100 m, the line losses were 20%. This confirms that DC deployments need to be studied on a case-by-case basis, as many parameters will influence the efficiency of the installation, such as the cabling distance, the final power, and the equipment quality.

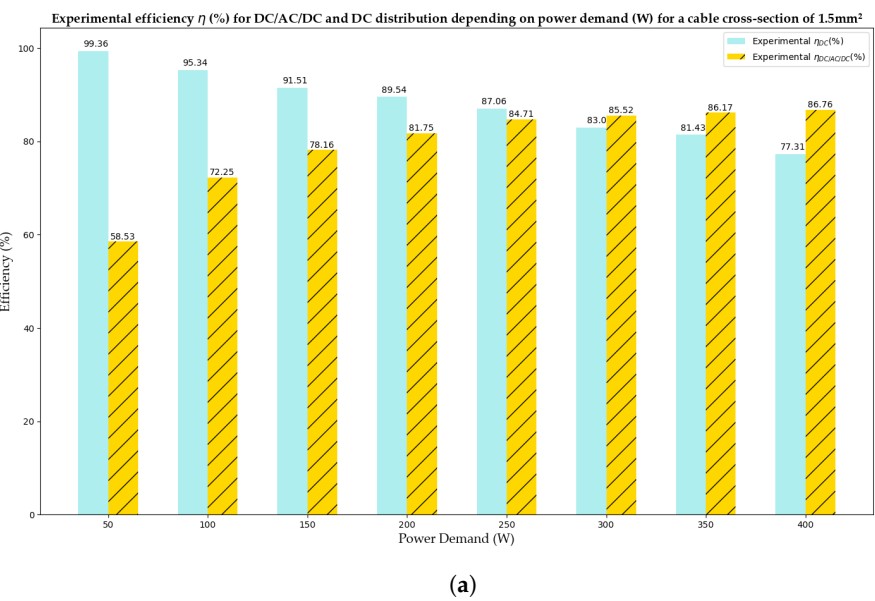

(**a**)

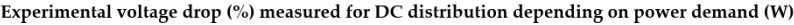

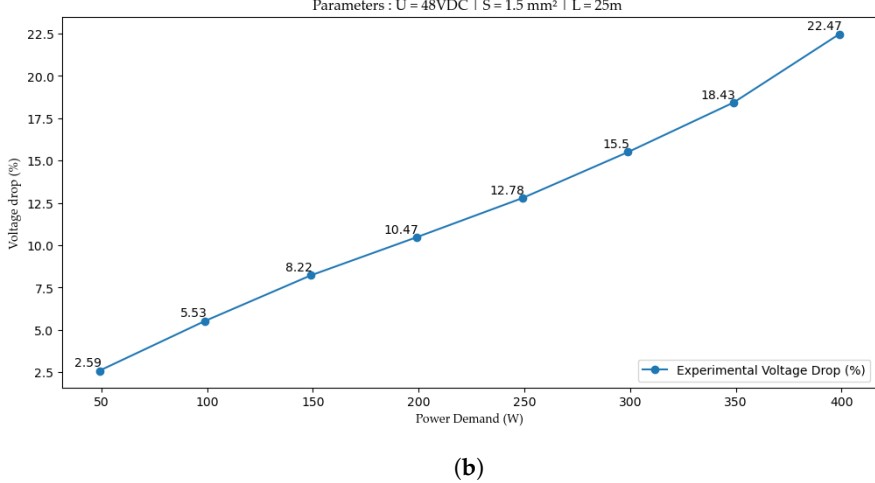

(**b**)

**Figure 14.** For a 1.5 mm$^2$ cable cross-section, comparison of efficiency $\eta$ (%) measured for DC and DC/AC/DC distribution (**a**), and evolution of experimental voltage drop (%) measured for DC distribution (**b**).

The discrepancy with the theoretical voltage drops calculated in Figure 9 can be explained by several factors, including the internal resistances of the battery and programmable load, the cable quality, the connector resistances, and the protection elements. This also means that special care must be taken during installation, and measurements must be carried out in addition to theoretical calculations. This additional check would verify that the voltage drops are respected and that the installation complies with current standards in order to avoid any safety risks for users and damage to terminal equipment.

In the case of the nanogrid deployment described in this paper, where the terminal loads are LEDs and DC fans with a maximum power demand of 40 W, the experimental efficiency was measured to be 41% more efficient than a conventional DC/AC/DC system.

### 4.3. Comparison between DC/AC/DC and 48 VDC Distribution for a Cable Cross-SECTION of 2.5 mm$^2$

A second comparison of the nanogrid efficiency for DC/AC/DC or DC distribution was made with a 2.5 mm$^2$ cross-section, and the results are shown in Figure 15a. As expected, DC efficiency is higher than for DC/AC/DC to around 400 W. At low power levels (below 50 W), the efficiency gain is not much higher than with 1.5 mm$^2$, so it may be more cost-effective to use 1.5 mm$^2$ for DC appliances, which do not require supply power superior to 50 W. This power value concerns appliances like lighting, televisions, rechargeable electronics, security systems, and some computers [41]. Furthermore, this result confirms that the cable cross-section necessary to use the IEEE 802.3 bt standard remains effective and there is no need to modify the installation parameters to gain efficiency, such as by using a section of cable higher.

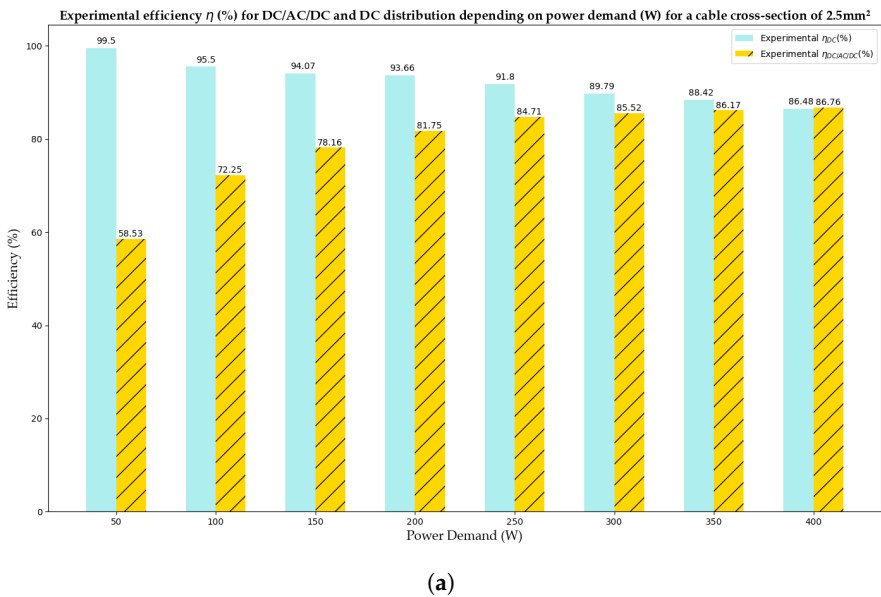

(**a**)

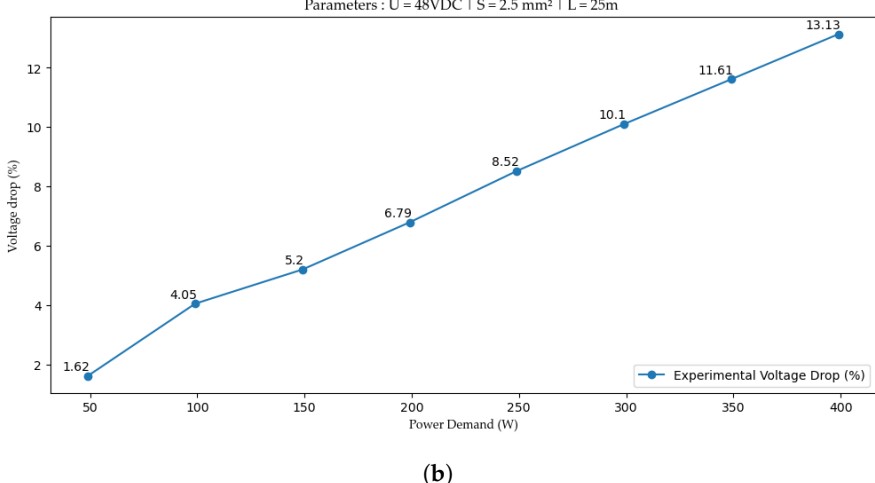

(**b**)

**Figure 15.** For a 2.5 mm$^2$ cable cross-section, comparison of efficiency $\eta$ (%) measured for DC and DC/AC/DC distribution (**a**), and evolution of experimental voltage drop (%) measured for DC distribution (**b**).

With respect to the voltage drops shown in Figure 15b, the same phenomenon was observed as with the 1.5 mm$^2$ section: the measured voltage drops were higher than expected. A voltage drop of 1.62% for 50 W up to 13% for 400 W was measured, instead of the theoretical 0.75% and 6% expected in (Figure 9). These results mean that although DC efficiency is higher than for DC/AC/DC to the value of 400 W, the power limit for a 25 m distance consistent with voltage drops at 5% would, therefore, be 150 W for this cable cross-section.

Similarly, additional complementary measurements could be included by considering a full operational cycle over a day for evaluation of the efficiency on a variable load profile. This profile would incorporate user movements as well as the solar energy variability.

These results indicate that it would be wise to combine higher voltage levels in DC buildings, such as 400 VDC, for example, in order to minimize the distances for 48 VDC distribution. Thus, combining 400 VDC and 48 VDC would enable a reduction in cable cross-sections for distribution distances exceeding 30 m within a DC-powered building. In close proximity to end-users, one option could involve the use of 400/48 DC converters to provide secure power supplies, ensuring voltages remain below 120 VDC, which is the threshold for "safety extra low voltage" in DC systems.

## 5. Conclusions

In this study, the deployment of an end-to-end 48 VDC nanogrid in a real-world enterprise is demonstrated and investigated. The nanogrid includes solar PV, LFP battery storage, and end devices powered via PoE 802.3 bt for a distance of 25 m between energy production and the final appliances. To quantify the efficiency and limits of the LVDC nanogrid compared to a conventional photovoltaic installation using 230 VAC, an experimental testbench is set up. Measurements are taken for two cable cross-sections (1.5 mm$^2$ and 2.5 mm$^2$) and are compared to the theoretical results calculated before the experiment.

For a 1.5 mm$^2$ cable cross-section, the efficiency of the DC nanogrid is measured to be between 40% and 23% higher than for a conventional DC/AC/DC installation, for final power loads of 50 W and 100 W, respectively. For a 2.5 mm$^2$ section, the efficiency of the DC nanogrid is measured to be between 40% and 16% higher than for a conventional DC/AC/DC installation, for a final power load in a range between 50 W and 150 W.

Despite the fact that the experimental results show that DC efficiency is higher than AC up to 250 W for 1.5 mm$^2$ and 400 W for 2.5 mm$^2$, the results for a maximum final power of 100 and 150 W are highlighted because experimental significant voltage drops were measured beyond these values. However, in order to produce DC installations with the same safety requirements as AC installations, it is necessary to comply with the NFC-15-100 standard, which stipulates that the voltage drop must not exceed 5%. As a comparison with the study [9], an energy gain of 6% was measured in a 12 VDC distribution compared to an AC distribution and it was specified that this could be improved with a higher voltage and terminal equipment having a suitable voltage, like 48 VDC, for example. Another similar study on measurements in a 48 VDC nanogrid [16] measured 30% higher efficiency compared to an AC system, but the distribution distance was not specified.

The investigation of the limits of the LVDC nanogrid carried out in this paper highlights that the efficiency measurement depends on several parameters which will be specific to the deployment conditions, namely, the wiring distances, the final power of the equipment required, the bus voltage chosen, and the efficiency of the converters.

These results are significant for potential energy savings in buildings. Indeed, over distances less than 100 m and low powers, it is confirmed that DC could optimize the electrical architectures. In the case of this work, the 48 VDC voltage was studied for its compatibility with numerous electrical devices and particularly the IEEE 802.3 bt standard. It can be used in innovative ways on devices other than IT equipment, such as lighting or DC fans.

A challenge lies in the fact that a standard DC bus voltage could be defined so that the DC terminal equipment is standardized to this voltage and can facilitate its installation.

To know the limitations of a DC compared to a DC/AC/DC installation in other deployment conditions, the novel methodology presented in this article can be applied elsewhere by changing the input parameters, such as the voltage, cable distance, cable cross-sections, and the converter efficiency. This methodology can, thus, be used as a design tool when deploying nanogrids.

It is suggested that the notion of costs could be integrated in order to obtain a technico-economic assessment in the building design phase. Finally, similar tests could be carried out with different cable qualities in order to check whether there would be quantitative improvements regarding voltage drops.

**Author Contributions:** Conceptualization, O.G., D.G.-C., F.A. and P.-O.L.d.P.; methodology, O.G.; software, O.G. and F.B.; validation, D.G.-C., F.A., P.-O.L.d.P. and J.-P.C.; formal analysis, O.G.; investigation, O.G.; resources, O.G.; writing—original draft preparation, O.G.; writing—review and editing, O.G., D.G.-C., F.A., P.-O.L.d.P. and J.-P.C.; visualization, O.G. and F.B.; supervision, D.G.-C., F.A., P.-O.L.d.P., L.L. and J.-P.C.; project administration, L.L. and J.-P.C. All authors have read and agreed to the published version of the manuscript.

**Funding:** This research is funded by the National Association of Research and Technology (ANRT), the company Intégrale Ingénierie, the University of La Réunion, and the University of Grenoble-Alpes.

**Data Availability Statement:** Data is contained within the article.

**Conflicts of Interest:** Authors Olivia Graillet, Flavien Bernard and Laurent Lemaitre were employed by the company Intégrale Ingénierie. The remaining authors declare that the research was conducted in the absence of any commercial or financial relationships that could be construed as a potential conflict of interest. The authors declare that this study received funding from Intégrale Ingénierie, the National Association of Research and Technology (ANRT), the University of La Réunion and the University of Grenoble-Alpes. Funders were not involved in the study design, collection, analysis, interpretation of data, the writing of this article or the decision to submit it for publication.

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
