# Peer review of "Optimizing Energy Consumption: A Case Study of LVDC Nanogrid Implementation in Tertiary Buildings on La Réunion Island"

_energies, doi:10.3390/en17051247_

Round 1

Reviewer 1 Report

Comments and Suggestions for Authors

The authors proposed the design, utilization and efficiency evaluation of a low voltage DC nanogrid. A realistic implementation for the designed LVDC nanogrid is achieved and applied in island of La Réunion. The nanogrid under study consisted of a PV system, a battery, and DC end-use equipment such as LED lighting and DC fans for two separate offices.                A system's performance comparison is carried out between the proposed nanogrid and a system uses a PV system with inverted voltage to 230 Vac.

The following remarks on the study are recorded:

1.       As you used different programs like VRM application and the SolarIO, I recommend to avoid missy illustration to provide a flowchart that simplifies your steps for capturing the data, analyzing it and providing the reference signals to the power flow system.

2.        I think that using the PoE is not a perfect choice as it has realistic disadvantages such as Limited power capacity, potential for network disruption and. potential for interference with data signal transmission. How you overcome these challenges?

3.       Cite the equations (3,4,5 and 6).

Author Response

Good morning, 

Thank you very much for your comments.

Please see the attachment which contains our answers.

Best regards,

Reviewer 2 Report

Comments and Suggestions for Authors

This article is well written, however, it would benefit from a few minor revisions. Increasing Efficiency in Tertiary Buildings through PoE LVDC Nanogrid in the island of La Réunion. Instead of relying on specific references such as "[17–20]" and "Cirque of Mafate [21,22]," it is more beneficial to provide a broader overview initially, focusing on the general principles of renewable Energy. Use up-to-date and pertinent research 2022-2024, as recommended In the following: 10.1016/j.est.2023.109289 ; 10.1016/j.renene.2022.11.006. This approach allows for a more comprehensive understanding of the topic before delving into specific details. By emphasizing the overall benefits and challenges associated with renewable Energy, the introduction can set the stage for a more nuanced discussion of its effects on local communities in remote regions.

2- Please review Relationship 2 for any potential errors or inconsistencies to ensure accuracy in conveying the intended message.

3- In the Conclusions section, write a paragraph comparing the current research with the studies reviewed in the introduction. Please submit a proposal for additional research. 10.1016/j.epsr.2023.110106

4- I think the title could be better. So I offer you 5 suggested improved titles:

1.       Enhancing Energy Efficiency in Tertiary Buildings: Deployment and Evaluation of a PoE LVDC Nanogrid in La Réunion

2.       Advancing Sustainability in Building Design: An LVDC Nanogrid Approach for Energy Efficiency in La Réunion

3.       Optimizing Energy Consumption: A Case Study of LVDC Nanogrid Implementation in Tertiary Buildings on La Réunion Island

4.       Power Over Ethernet Innovations: Improving Energy Efficiency with LVDC Nanogrids in Insulated Climates

5.       Towards Sustainable Energy Solutions: Assessing the Impact of LVDC Nanogrids on Building Efficiency in La Réunion

5- Abstract Review and Suggestions for Improvement:

  1. The abstract provides a clear overview of the study on implementing a PoE LVDC nanogrid in La Réunion to enhance energy efficiency in tertiary buildings. However, there are areas for improvement to enhance clarity and precision.
  2. The abstract can benefit from providing more specific details on the methodology employed for efficiency quantification. This would help readers understand the process and ensure transparency in the evaluation.
  3. The abstract mentions that the efficiency of the nanogrid is measured to be 40% higher than an installation powered by PV distributed at 230 VAC. It would be beneficial to include specific efficiency metrics or calculations to support this claim and provide a clearer comparison.
  4. While the abstract highlights the components of the nanogrid and the use of PoE for energy management, more emphasis on the unique contributions or innovations of the study would enhance its impact and relevance to the field.
  5. Consider incorporating key findings or implications of the study in the abstract to provide readers with a glimpse of the significance of the research and its potential implications for sustainable energy solutions in insulated climates like La Réunion.
Comments on the Quality of English Language

This article is well written, however, it would benefit from a few minor revisions.

Author Response

Good morning, 

Thank you very much for your comments.

Please see the attachment which contain our answers.

Best regards,

Reviewer 3 Report

Comments and Suggestions for Authors

The author investigates the power efficiency of PoE LVDC Nanogrid for tertiary buildings. The main comments are summarized below:

1) In the literature review sector, the main drawbacks of existing research on the design of LVDC nano grid should be highlighted, and how the trackbacks are considered/addressed by the proposed method should be added.

2) In section 2.1 on hardware architecture, the operation modes and the corresponding power flow conditions of the proposed LVDC nanogrid should be provided.

3) The energy management strategy of the proposed LVDC nanogrid should be provided.

4) In the results and discussion section, it is suggested that the real operation scenario of an LVDC nanogrid can be used to perform the efficiency evaluation.

Instead of evaluating the power efficiency at a specific power, a weighted efficiency can be considered for evaluating the efficiency over a complete power cycle (similar to the California Energy Commission efficiency).

Author Response

(The authors gave the same response as above.)

Round 2

Reviewer 1 Report

Comments and Suggestions for Authors

The revised version is improved. The authors provided sufficient replies to the raised criticisms. I recommend with the acceptance.

Author Response

Good morning,

Thank you for your feedback and validation of our article submission.

Best regards,

Reviewer 2 Report

Comments and Suggestions for Authors

The article has been well-edited and is now ready for submission for publication.

Comments on the Quality of English Language

The article has been well-edited and is now ready for submission for publication.

Author Response

(The authors gave the same response as above.)

Reviewer 3 Report

Comments and Suggestions for Authors

Thank you for submitting the revised manuscript. The majority of the concerns have been dealt with. Below are my additional comments:

1. About comment 4, even though the measurements over a complete day are inaccessible, it is recommended that the authors utilize the emulated typical operating profiles of an LVDC Nano-grid for the efficiency assessment. This is essential as the results from the case studies show that efficiency is power-dependent. The CEC efficiency or European efficiency is commonly used to assess the efficiency of a power inverter over its entire power cycle. Although these two measurements may not be suitable for LVDC nanogrid, the fundamental principles can be referenced.

Link for the CEC efficiency and European efficiency:

https://www.solarchoice.net.au/blog/types-of-solar-inverter-efficiency/

Author Response

Good morning.

Thank you very much for the precisions, it is very interesting for improving our work. We added the following paragraph in subsection "3.2.1. Efficiency for DC/AC/DC Distribution" :

"In the study of DC/AC efficiency, particularly concerning inverter, formulas and testing protocols have been developed in the United States by the California Energy Comission (CEC) and Sandia National Laboratories \cite{ref15_CEC, ref15_sandia_lab}. In Europe, the same work adapted to the local climate was carried out by by the Joint Research Centre (JRC), the scientific and technical research laboratory of the European Union \cite{ref15_jrc_eu}. Their research has notably determined weighting coefficients to be applied across various operating levels of an inverter throughout its entire power cycle. While not utilized within the scope of this study, it is planned to integrate this methodology in future work."

Best regards